# Narrative Review: Update on the Molecular Diagnosis of Fragile X Syndrome

**DOI:** 10.3390/ijms24119206

**Published:** 2023-05-24

**Authors:** Cristian-Gabriel Ciobanu, Irina Nucă, Roxana Popescu, Lucian-Mihai Antoci, Lavinia Caba, Anca Viorica Ivanov, Karina-Alexandra Cojocaru, Cristina Rusu, Cosmin-Teodor Mihai, Monica-Cristina Pânzaru

**Affiliations:** 1Medical Genetics Department, Faculty of Medicine, “Grigore T. Popa” University of Medicine and Pharmacy, University Street No 16, 700115 Iasi, Romania; ciobanucristian@yahoo.com (C.-G.C.);; 2Investigatii Medicale Praxis, St. Moara de Vant No 35, 700376 Iasi, Romania; 3Medical Genetics Department, “Saint Mary” Emergency Children’s Hospital, St. Vasile Lupu No 62, 700309 Iasi, Romania; 4Pediatrics Department, “Grigore T. Popa” University of Medicine and Pharmacy, University Street No 16, 700115 Iasi, Romania; 5Department of Biochemistry, Faculty of Dental Medicine, “Grigore T. Popa” University of Medicine and Pharmacy, University Street No 16, 700115 Iasi, Romania

**Keywords:** fragile X syndrome, long read, methylation, PacBio sequencing, Oxford Nanoporesequencing

## Abstract

The diagnosis and management of fragile X syndrome (FXS) have significantly improved in the last three decades, although the current diagnostic techniques are not yet able to precisely identify the number of repeats, methylation status, level of mosaicism, and/or the presence of AGG interruptions. A high number of repeats (>200) in the fragile X messenger ribonucleoprotein 1 gene (*FMR1*) results in hypermethylation of promoter and gene silencing. The actual molecular diagnosis is performed using a Southern blot, TP-PCR (Triplet-Repeat PCR), MS-PCR (Methylation-Specific PCR), and MS-MLPA (Methylation-Specific MLPA) with some limitations, with multiple assays being necessary to completely characterise a patient with FXS. The actual gold standard diagnosis uses Southern blot; however, it cannot accurately characterise all cases. Optical genome mapping is a new technology that has also been developed to approach the diagnosis of fragile X syndrome. Long-range sequencing represented by PacBio and Oxford Nanopore has the potential to replace the actual diagnosis and offers a complete characterization of molecular profiles in a single test. The new technologies have improved the diagnosis of fragile X syndrome and revealed unknown aberrations, but they are a long way from being used routinely in clinical practice.

## 1. Introduction

Fragile X syndrome (FXS) (OMIM 300624) is the most common inherited form of intellectual disability and the leading monogenic cause of autism spectrum disorder worldwide. It is due to an expansion of the triplet cytosine–guanine–guanine (CGG) within the fragile X messenger ribonucleoprotein 1 gene (*FMR1*) [1,2,3].

The syndrome is characterized by developmental delay, behavioural problems, facial dysmorphism (macrocephaly, long narrow face, prognathism, prominent and large ears), and macroorchidism (usually postpubertal). Hypotonia, seizures, recurrent otitis media, sleep disorders, strabismus, gastroesophageal reflux, joint laxity, pectus excavatum, pes planus, soft skin, mitral valve prolapse and aortic root dilatation are cited less [4,5,6]. The neuropsychiatric characteristics in FXS patients include autism, hyperactivity, hand flapping and hand biting, anxiety, poor eye contact, phobias, restricted interests, social and communication deficits, tactile defensiveness, self-injurious behaviour, and aggressiveness [4,7]. Phenotype severity is variable and depends on molecular aspects (level of methylation and mosaicism), genetic modifiers, and environmental factors [8,9]. The features described in males with FXS have also been reported in females heterozygous for the fragile X messenger ribonucleoprotein 1 gene (*FMR1*; OMIM 309550) full mutation (FM) but with a lower frequency and milder involvement (borderline IQ and learning and emotional problems). The phenotype in females is correlated with the X inactivation pattern [5].

This article aims to describe the most important techniques used for fragile X syndrome diagnosis, including the limitations and advantages of each, in order to improve diagnosis and management.

## 2. Genetics, Epidemiology, and Aetiology

Most (>99%) of the pathogenic variants are due to the expansion of the CGG trinucleotide repeat (TR) (>200) in the promoter region of *FMR1* [9,10]. According to the number of repeats, *FMR1* alleles are classified as normal with 5–44 TR, intermediate with 45–54 TR, premutation (PM) with 55–200 trinucleotide repeats (TRs), and FM with over 200 TRs [11,12].

The prevalence of FM ranges from 1 in 4000 to 1 in 5000 in males and 1 in 4000 to 1 in 8000 in females. The variability in prevalence rates of FM across studies may be explained by sample selection, founder effect, or ethnic differences. The prevalence of PM is estimated for males as 1 in 250 to 1 in 850 and for females as 1 in 110 to 1 in 300 [13,14]. The use of different detection methods, imprecision in laboratory measurements of repeat numbers, and nonidentical definition of PM could contribute to the variations in prevalence [12].

FXS is determined by mutations in the *FMR1* gene (fragile X messenger ribonucleoprotein 1 gene; OMIM 309550), located on Xq27.3, where the folate-sensitive fragile site FRAXA was initially described in affected males. It contains 17 exons, encodes for protein FMRP, and undergoes alternative splicing resulting in different isoforms.

Four epigenetic factors regulate *FMR1* gene transcription, the most important one being cytosine methylation [15], which blocks *FMR1* gene transcription.

The promoter of the *FMR1* gene contains CpG dinucleotides, sites for transcription factors, GC-boxes, initiator-like sequences, and a CGG repeat usually with AGG interruption (after 9–10 CGG repeats) [15,16]. Normal alleles have an upstream methylated region and a downstream unmethylated region with a distinct DNA-methylation boundary (650–800 nucleotides upstream of the CGG repeat). The mechanism of FMR1 gene inactivation in FM is the loss of the normal methylation pattern due to hypermethylation of cytosines within CGG repeats and CpG islands.

FXS cases in mosaic (some cells with FM, others with PM or cells containing methylated *FMR1*, and others with unmethylated *FMR1*) have a milder phenotype. Patients with deletions or point mutations have also been reported. PM alleles are unstable, and factors that increase the risk of expansion to FM include maternal transmission, an increased number of TRs, and the absence of AGG interruptions within the repeat [13,17,18,19].

PM carriers are at increased risk for a spectrum of conditions: fragile X-associated tremor/ataxia syndrome (FXTAS), fragile X-associated primary ovarian insufficiency (FXPOI, OMIM 311360), fragile X-associated neuropsychiatric conditions (FXANC), and other anomalies (autoimmune disorders, fibromyalgia, and thyroid dysfunctions) [20]. FXTAS is a late-onset neurodegenerative disorder characterized by progressive intention tremor, cerebellar ataxia, neuropathy, autonomic dysfunction, parkinsonism, and cognitive decline [9,21,22]. FXPOI is defined as hypergonadotropic hypogonadism and oligo/amenorrhea before 40 years of age. FXANC include anxiety, depression, social communication deficits, and attention problems [22,23].

The idea of a role for AGG interruptions was first promoted by Eichler et al. [24]. The interruptions are usually present at the 5′ ends of the repeat tract. Usually, they appear after a periodicity of 9–10 CGG repeats. The number of interruptions is significatively lower for patients with a positive family history of FXS.

Fu et al. were the first who brought the idea that nonrepeat elements can contribute to a higher risk [16,25]. The technical limitations could not provide the answer to why smaller numbers of repeats were transmitted as FM, and the higher numbers of repeats were not transmitted as FM [25].

Yrigollen et al. concluded that an allele with 75 tandem repeats and no interruptions has a 77% chance of becoming FM, but only a 12% chance of becoming FM if it has AGG interruptions [26]. Nolin et al. discovered that the loss of AGG interruptions happens after contraction events of a maternal PM, and the resulting sequence is highly susceptible to becoming an FM allele [18].

Another important aspect observed in FXS refers to somatic mosaicism. The cells from the same patient can have different numbers of CGG repeats and different methylation profiles [27]. Cells can enter mitosis without completing the replication of the *FMR1* region, in the presence of folate deprivation [28]. The late replication of FM alleles can explain the large number of CGG repeats, which can form secondary structures called hairpins that delay the progression of replication. A large number of repeats can also form other secondary structures, triplex or quadruplex, which can be the key to instability mechanisms seen in these disorders [19]. The quadruplex results from a single-stranded CGG repeat, stabilized by a G quartet, where four guanines are linked by Hoogsteen hydrogen, which is a bond between a purine base and a pyrimidine base in a way that allows the formation of an additional hydrogen bond between a hydrogen atom located on purine base and a nitrogen atom from the pyrimidine base [29]. The quadruplex is a stable structure, is most frequently found in the *FMR1* coding strand, and has multiple repeats [30]. The AGG interruptions interfere with the quadruplex and decrease the stability of this structure [31].

FMRP is an RNA-binding protein with a central role in the translational regulation of a large number of mRNAs, many of which are involved in synaptic development and plasticity. It is expressed almost ubiquitously at low levels, but the highest concentrations are found in the brain and testis. FMRP is involved in mRNA transport (nucleus to cytoplasm export), stability, initiation of translation, ribosomal translation, regulation of chromatin modulators, and RNA editing. The absence of FMRP dysregulates glutamatergic and GABA signalling, the endocannabinoid system, and the bone morphogenetic protein receptor 2 (BMPR2)—cofilin pathway involved in numerous neuronal processes [32,33,34]. FMRP also modulates the activity of many ion channels: voltage-dependent ion channels (sodium, potassium, calcium, and hyperpolarization-activated cyclic nucleotide-gated channels), voltage-independent ion channels (small conductance Ca^2+^-activated K^+^ channels) and ligand-gated ion channels, which contribute to abnormal excitability [10]. FMRP is also involved in genome stability at the chromatin level in DNA damage response [35,36].

## 3. Cytogenetic and Molecular Diagnosis for Fragile X Syndrome

We describe below, the most important techniques used for FXS diagnosis and when they were used for the first time (Figure 1).

### 3.1. Karyotyping

The *FMR1* gene was identified in 1991. Before this, the only diagnostic test was the karyotype analysis (Figure 1). The fragile site is located on Xq27.3 [37] and becomes evident in special cell culture conditions (cell culture medium deprived of folic acid). The major disadvantages refer to the difficulty in identifying the site and the necessity to read approximately 100 metaphases because the fragile site is not present in every metaphase, which is the reason why this technique is no longer used (Table 1).

### 3.2. Southern Blot

Southern blot, named after British molecular biologist Edwin Southern, is a technique in which purified DNA from a biological sample is digested with restriction enzymes. The resulting DNA fragments are separated by using an electric field to move them through a gel, allowing fragments to migrate depending on their length. Transferring the DNA fragments from the gel or matrix onto a solid membrane allows them to be exposed to a DNA probe that has been marked with a radioactive, fluorescent, or chemical tag. The tag enables the Southern blot to display any DNA fragments with complementary sequences to the DNA probe sequence (Figure 2) [38].

Southern blot is cited as the gold standard for FXS diagnosis. It has multiple advantages and can detect *FMR1* alleles, with normal size repeats, PM and FM. It can also reveal the methylation status of the *FMR1* promoter region. The limitations of this technique are that it is not capable of appreciating the exact number of repetitions, and it needs a large amount of genomic DNA as it is not able to identify deletions or point mutations [4,39] (Table 1).

Baker et al. analysed methylation status in 87 male patients using methylation-sensitive Southern blot (mSB) and methylation-specific quantitative melt analysis (MS-QMA), which is a technique used to identify the *FMR1* methylation exon 1/intron 1 boundary. The patients are diagnosed with FXS mosaicism. Furthermore, they tried to correlate *FMR1* methylation in blood and buccal epithelial cells with intellectual disability. The results obtained with MS-QMA were superior, and the methylation analysis was correlated with intellectual disability in all analysed tissues. The results obtained after mSB did not show an association between methylation in blood and intellectual disability. They demonstrated incomplete silencing of *FMR1* in the patients with methylation mosaicism, which showed that MS-QMA gave superior results to mSB [40].

### 3.3. Triplet Repeat Primed PCR (TP-PCR)

Triplet repeat primed PCR (TP-PCR) is a technique developed initially to evaluate the expanded alleles in myotonic dystrophy. This technique uses a specific primer flanking the repeat region, upstream or downstream, and a triplet-primed primer with a 3′ sequence, which hybridizes within the repeat region and generates multiple amplicons. The amplicons are size different by a repeat unit. The last one is a tail primer with the same sequence as the 5′ overhanging sequence of the first primer, which increases further amplification of the generated fragments by the first primer and the extension of the first primer. The expansions generate a heterogeneous mixture of TP-PCR amplicon fragments [41,42].

Triplet repeat primed PCR has multiple applications, such as Friedreich Ataxia [43], myotonic dystrophy [44], Huntington’s disease [45], FXS, and short tandem repeat diseases. For FXS, this technique can identify the FM, can differentiate normal and FM females, and it can offer information about the size of the CGG repetition number [46]. The most significant limitations of this method include its failure to determine the PM carriers, quantify the precise number of repeats, and evaluate the methylation status (Table 1) [47].

Curtis-Cioffi et al. compared PCR-based techniques with Southern blot by analysing the same samples with both techniques. They performed one PCR reaction and then one more PCR for the probes that were not amplified after the first reaction. The samples were analysed by Southern blot. The results were not entirely coincidental, since for 5 samples out of the 75 analysed, the results were not the same, with differences being recorded in the case of FM and PM. The Southern blot method was more reliable in quantifying the number of repeats than the PCR approach [48].

Rajan-Babu et al. described a screening method for FXS using two direct TP-PCRs. First, TP-PCR analyses the melting of the amplicons, which will reveal the PM and FM. Melting curve analysis triplet repeat PCR will reveal the mosaicism too, but it is unable to discriminate between PM and FM samples. For the probes that are positive after the first assay, another PCR will be performed. The second TP-PCR is coupled with capillary electrophoresis. Capillary electrophoresis TP-PCR is capable of fully characterising the repeat number and identifying the presence of AGG interruptions. The main concern about the method is the possibility of false-positive results if the probe is contaminated [49].

### 3.4. Digital PCR

Digital PCR (d-PCR) is classified as third-generation PCR. The technique combines the classic PCR reaction with fluorescence-based detection, usually used in real-time quantitative PCR [50]. The DNA is divided into numerous small volume compartments in which the molecules are distributed. After amplification, the absorbance is measured in every compartment. A reaction with no target molecules gets a 0, and a reaction that has one target molecule receives a 1. The copy number and the density are calculated using Poisson statistics and the number of PCR positive reactions. In most cases, one compartment can contain more target molecules, and the result should be corrected using Poisson statistics [51].

A variation of Digital PCR is Digital Droplet PCR (ddPCR). It is based on a water–oil emulsion. The DNA is randomly subdivided into water-in-oil droplets and is independently amplified. For the detection, a two-colour optical system is used. The droplets with similar volumes are selected for fluorescence detection [51].

D-PCR has multiple advantages, including the fact that it enables quantification [52], it does not require a standard curve [53], it has high precision, accuracy, and sensitivity [54], and the PCR bias is minimal. The main limitations of the technique are the narrow dynamic range and the relatively high costs. (Table 1).

**Table 1 ijms-24-09206-t001:** Diagnostic techniques for Fragile X Syndrome: Applications.

	TRN	MS	PM	AGG Int	MOS	Del Ins	P. mut	Advantages	Disadvantages
**Karyotype**	–	–	–	–	+	±	–	Cheap.	Outdated. Inaccurate. Faulty.
**Southern blot**	±	+	±	–	+	+	–	Golden standard. (repeat expansion and methylation status) [55].	Very labour-intensive. Time-consuming. Not in routine diagnostic settings [55,56].
**TP-PCR**	+	–	+	±	±	–	–	Rapid. Facile. Routine diagnostic. High sample throughput [43].	Does not provide the size of expanded CGG repeats.
**D-PCR**	+	–	+	–	+	+	–	Low costs. High number of fragments.	Costs with equipment and consumables.
**MS-PCR**	–	+	–	–	±	–	–	High sensitivity and specificity.	False-positive results.
**MS-MLPA**	–	±	–	–	±	±	±	Cheap. Golden standard (copy number).	False-positive results. Not for females.
**Illumina seq**	±	–	+	+	±	+	+	Standardized technique. Ideal for point mutations.	Not for large expansions.
**PacBio seq**	+	+	+	+	+	+	+	High coverage and accuracy [57]. A single assay.	High costs for equipment.
**Nanopore seq**	+	+	+	+	+	+	+	Scalable (Flongle for few patients and PromethION for large numbers). Time and space efficient [58]. All in a single assay.	Only for research use. Complex bioinformatics interpretation.
**OGM**	±	–	–	–	+	+	–	High-resolution genome-wide analysis of all structural variants [59].	Low throughput. Not for PM. Only for research use.

+ The technique is suitable and recommended for diagnosis; ± the technique is suitable but error-prone; – the technique is not recommended for diagnosis. TRN = tandem repeats number; MS = methylation status; PM = premutation; AGG Int = AGG interruption; MOS = mosaicism; and P. mut = point mutation.

Third-generation PCR has multiple applications. Some examples are viral quantification [60], oncology and haematology [61], and noninvasive prenatal diagnosis (NIPD) [62].

Alvarez-Mora et al. attempted to correlate the occurrence of FXPOI in PM carriers with the expression profile of three long noncoding RNAs (lnc_RNAs) derived from the FMR1 gene locus, *FMR4*, *FMR5,* and *FMR6*. Lnc_RNAs are untranslated RNA molecules of more than 200 nucleotides in length that play an important role in gene expression. They extracted RNA from 36 PM carriers, 20 were diagnosed with FXPOI and 16 without FXPOI. Afterwards, they performed reverse transcription. They used ddPCR to quantify the expression. The results were only significant for *FMR4*, and the PM carriers with FXPOI had higher expression levels, which can be used as a biomarker for screening [63].

One of the main advantages of ddPCR is the possibility of characterizing low-level mosaicism, including FXS. Digital droplet PCR can accurately quantify the number of short tandem repeats in the *FMR1* gene, but it is unable to characterize the methylation status (Table 1) [64].

### 3.5. Methylation-Specific PCR

For the methylation-specific PCR (MS-PCR) technique, the DNA goes through a process of bisulphite conversion, which converts all unmethylated cytosine to uracil. Two pairs of primers are used: one for methylated DNA and one for unmethylated DNA. For discrimination, at least one CpG site is included in each primer sequence. Using specific primers for the methylated DNA, successful amplification indicates that the amplified region is methylated [65].

The main advantages of this method are the high sensitivity and specificity to reveal the methylation status (Table 1). The drawbacks of this technique are the incomplete conversion of cytosine to uracil [66], the false-positive results, and the time-consuming process [67].

Berry-Kravis et al. performed FXS and other *FMR1*-related disorder diagnoses and screening using AmplideX Fragile X Dx and the Carrier Screen Kit, which includes two PCR reactions (one TR-PCR and one MS-PCR). The methylation could be observed with better precision than the Southern blot method by using a much more specific PCR in cases where the methylation percentage was over 20%. By the same approach, they were also able to highlight the AGG interruptions with higher precision in the case of male patients (by counting peaks and identifying the spaces corresponding to AGG repeats). In the case of female patients, it lacks precision regarding the localization of the specific allele carrying the interruptions; however, it is still able to reveal how many AGG interruptions are present [68]. Furthermore, they accurately quantified the number of repeats, including the high or intermediary values. The technique can quantify the mosaicism too, as a low-level peak, depending on mosaicism frequency, but it can also have false-positive results [69].

### 3.6. Methylation-Specific Multiplex Ligation-Dependent Probe Amplification

Methylation-specific multiplex ligation-dependent probe amplification (MS-MLPA) can identify the methylation status and the copy number variation. The technical difference between MLPA and MS-MLPA is the necessity of an additional reaction for the methylation profiling in which a restriction enzyme HhaI is added to digest unmethylated DNA; otherwise, if DNA is methylated, digestion is inhibited. One reaction is performed with only the ligase, the other one with the restriction enzyme and the ligase. The undigested reaction is used for copy number variation estimation and the reaction with digestion is used for establishing methylation status [70].

Gatta et al. conducted a retro-prospective study using MS-MLPA ME029. They retrospectively analysed 28 cases (23 males and 5 females) with FM, 2 with PM, and 21 normal controls. Prospectively, they analysed 119 patients of which 98 were intellectually disabled males, one was a male fetus, and 20 were females (7 with FM, 5 PM, and 8 normal controls). For all of the patients with FM, MS-MLPA showed the presence of hypermethylation of the *FMR1* promoter, which was confirmed by other techniques. The results for the female patients could not be analysed due to the presence of 2 X-chromosomes. They also showed the detection limit for MS-MLPA. They made serial dilutions to 2.5%, whereupon the results became nonreproducible. The methylated mutation must represent at least 5% of the total DNA to be visible on MS-MLPA [71].

While CGG expansion’s repetitive nature makes it challenging to find deleted alleles in this region, MS-MLPA has the benefit of being effective at evaluating CNVs along the *FMR1* gene in addition to the methylation status. The main limitations of MS-MLPA relate to the impossibility of detecting inversions or translocations within the gene and the fact that it cannot count the number of CGG repeats. (Table 1)

### 3.7. Optical Genome Mapping

Optical genome mapping relies on the capability of the Saphyr system to directly visualize ultra-long labelled DNA molecules. Ultra-high-molecular-weight DNA with an average length of 200 kbp is fluorescently labelled at the level of a specific motif sequence of 6bp (CTTAAG), which occurs approximately every 5 kbp in the human genome. After that, it is linearized in a nanochannel array (Saphyr chip) and visualized by the Saphyr system. These labels generate a specific pattern along the DNA molecules, which allows them to be mapped to a particular region in the reference genome. The generation of these maps allows the identification of most structural and numerical anomalies, as well as repeat expansions or contractions (Table 1) [72,73].

For analysis of repeat disorders, Bionano Genomics developed two targeted workflows, EnFocus FSHD, which measures repeat contractions in facioscapulohumeral muscular dystrophy, and EnFocus fragile X analysis, which measures repeat expansion in the *FMR1* gene. Regarding the predictivity and analytical sensitivity of the EnFocus Fragile X analysis, they were established at 100% and 97%, respectively [74].

A multisite study by Iqbal et al. compared the analysis of structural variants using Optical genome mapping (OGM) with the current standard of care represented by chromosomal microarray, karyotyping, fluorescence in in situ hybridization, Southern blot analysis, and PCR, in postnatal constitutional cases. The authors analysed 404 samples using the Bionano EnFocusTM Fragile X Analysis workflow. A total of 401 of them were able to be classified as either full expansion or not full expansion. The remaining 3 samples were classified as inconclusive, the standard of care indicating PM or full expansion close to the 200 repeats threshold. In total, 93 samples representing 33 unique cases were classified as positive for *FMR1* full expansion; these results showed 100% concordance with measurements obtained by the standard of care (Southern blot analysis and PCR) [75]. One caveat of using OGM for testing *FMR1* repeat expansion is that OGM is not suited for testing for PM or when the number of repeats is near the threshold of 200, in which case, other tests such as Southern blot or PCR-based techniques may be needed (Table 1). An expansion above 220 repeats should be considered pathogenic [76].

### 3.8. Short-Read Sequencing

Short-read sequencing or second-generation sequencing methods are divided into two categories: sequencing by hybridization and sequencing by synthesis.

Most of the sequencing by synthesis uses a method in which the DNA strands are separated in millions of wells or fixed in specific locations. The DNA is amplified and then begins the synthesis reaction in which the labelled nucleotides can be identified [77].

The most popular example of sequencing by synthesis is Illumina, which uses “bridge amplification”. Approximately 500 bp DNA strands with the adapters on both ends are used to proceed with repeated amplification reactions on a solid support. The support contains oligonucleotides complementary to a ligated adapter. There is a space between the oligonucleotides from the slide, which allows the amplified DNA to create clonal “clusters” composed of 1000 copies of each oligonucleotide fragment. Many parallel clusters occur on each glass. The four bases have different fluorescent labels. The nucleotides are incorporated and identified during the synthesis reaction. The nucleotides also serve as synthesis terminators for each reaction, which are unblocked following detection for the subsequent round of synthesis [78].

The advantage of this approach to the diagnosis of fragile X syndrome is the capacity to resolve rare cases without a high number of STR. The drawbacks are represented by the difficulty in correctly mapping the STR to the reference to exactly quantify the number of them, and its inability to reveal the methylation status (Table 1).

Sitzmann et al. presented the case of a 10-year-old child with a highly suggestive phenotype for FXS. The number of repetitions identified was 24, which is within normal ranges. An array was also performed, but the result was not fully correlated with the phenotype. The diagnosis was clarified after a gene panel, performed on short-read sequencing, which showed the presence of hemizygous guanine to adenine transition (c.413G>A) in *FMR1* [79]. In the literature, other point mutations are reported, some in promoter regions [80], or missense mutations present in the *FMR1* gene [81]. This kind of change can easily be revealed on short-read sequencing and can clarify the diagnosis.

### 3.9. Long-Range Sequencing

Long-range sequencing is represented by two technologies—PacBio sequencing and Oxford Nanopore sequencing.

PacBio sequencing is a method for real-time sequencing [82]. The information is captured during the replication process and the template is called SMRT-bell. This is a single-stranded circular DNA strand, which results from the double-stranded DNA target molecule. The target molecule has two hairpin adaptors, one at each end. The DNA strands are ligated on both ends to form a circular DNA molecule [83]. When the template is loaded, it diffuses in a room called a zero-mode waveguide. Here, a polymerase will start the replication from one end. The labelled nucleotides emit different light spectra. All of the images are collected by a “movie”, and all of the impulses from a zero-mode waveguide are a single nucleotide sequence (continuous long read) [84]. The DNA polymerase completes multiple circles of the same DNA molecule in a single read, which increases the coverage (Figure 3) [85].

The pace of the polymerase progressing through the DNA strand is registered during sequencing. The time between nucleotide incorporations is called the interpulse duration (IPD) and it is different because of the methylation of the DNA. A methyl group on one nucleotide will affect the incorporation rate of the other nucleotides and can reveal the methylation status [86].

PacBio sequencing allows the diagnosis of the FXS with amplification or amplification free. The region is low complexity, which makes it difficult to amplify. For heterozygous patients with a normal allele and one allele with expansion, it is only possible to amplify the normal allele [87]. Liang et al. characterized the *FMR1* locus with PacBio long-range sequencing after an amplification. They studied 62 patients. The method quantified the repeats (93 to 940), identifying the complete mutations in 30 patients. They also identified both the interruptions and the mosaicism. Two patients had some rare variants, a deletion of 237 kb and a deletion of 774 kb. The study concluded that the method of long-read sequencing is 2–4 times more sensitive than TP-PCR [88].

Another possibility is to directly sequence the genomic DNA. Tsai et al. used an amplification-free abort. They digested the genomic DNA with two restriction enzymes, ECoRI-HF and BamHI-HF, and cut the SMRT-bell template, which contains the region of interest with Cas9 (Crispr-associated protein) and cr-RNA complex, which is complementary to a region near the region of interest. A new adapter was ligated to the digested template, which had a role in purification with MagBead, to enrich the region of interest. After this, the DNA was sequenced with PacBio SMRT sequencing. The digestion step had the role of increasing the on-target reads from 2% to 9%. The advantages of the method are that it can quantify the number of repeats for FXS, can detect interruptions, and the study revealed the possibility of targeting multiple regions. The main limitation of this method is genetic variations. The presence of polymorphisms can guide the cas9 complex to cut nonspecifically [89].

Oxford Nanopore technology is another long-read sequencing technology. Nanopore sequencing can directly sequence the nucleotides without synthesis. The technology relies on the detection of the changes in the ionic current generated by the crossing of a strand of DNA or RNA through a protein nanopore, stabilized in an electrically resistant polymer. It can also discriminate between methylated and unmethylated cytosine [90,91]. Oxford Nanopore technology has multiple approaches in diagnosing FXS, also with amplification and amplification free.

Payne et al. developed an alternative approach, “Selective sequencing” or “Read until”. Leveraging the potential of Nanopore sequencers to sequence in real time, it is possible to selectively sequence the DNA molecules by reversing the voltage across individual nanopores, which will specifically reject certain molecules of DNA. This results in better coverage and more on-target reading. Furthermore, the target selection is flexible and does not require any supplementary laboratory materials [92].

Stevanovski et al. used a custom panel comprising 37 STR loci, including *FMR1*, and performed selective sequencing. At the STR site in the *FMR1* 5′ untranslated region, they obtained a median of 19 coverage for women and 9 for men. They were able to quantify the number of repetitions (between 20–654 copies). The technique discriminated between healthy, PM, and affected patients. Furthermore, the technique identified the AGG interruption. They also sequenced *FMR2*, and the results were classified as normal. Another advantage of this technique is the possibility of methylation profile detection. In this study, the results revealed hypermethylation in patients with FM (>75% median methylation). The subjects with PM or normal results had a low methylation frequency, except for one woman with PM. This woman had a differential methylation between the two *FMR1* alleles: the PM predominant was hypermethylated and the normal allele was unmethylated. The advantages of this approach are the possibility of adding other targets and the possibility of seeing epigenetic changes. The main limitation is the modest on-target coverage [93].

Giesselmann et al. tried a different approach using Cas cleavage and sequencing on Nanopore. They analysed two probes with *FMR1*-distinct repeat expansion. They also developed an algorithm called STRique (short tandem repeat identification, quantification and evaluation) to quantify the number of repeats. The software aligns the conventional base-called sequence to the reference. After this, the software maps the limits of each repeat upstream and downstream. Finally, STRique quantifies the number of repeats. To improve the coverage for *FMR1*, they used Crispr-Cas12-a (clustered regularly interspaced short palindromic repeats), RNP, and Crispr-Cas9 in parallel to cut the DNA. Cas9 sequencing increased sequencing depth with a reduction for off-target reads. Then, they evaluated the methylation profile and observed the expanded alleles that were fully methylated. The results were confirmed with other techniques [94].

Zhou et al. sequenced the locus for *FMR1* using the Cas9 approach (Figure 4), quantifying the repetitions and focusing on distal methylation. They suggested that the silencing of *FMR1* could be the result of long-range mechanisms beyond local DNA methylation. They investigated how the chromatin architecture and epigenetic modifications are altered throughout the genome as a consequence of a CGG tract length. They analysed pluripotent stem cell lines differentiated to neural progenitor cells, with a variable number of CGG repeats, from normal to FM. They cut the DNA around the transcription start site of the *FMR1* gene with Cas9 and then quantified the repetitions with Nanopore sequencing. The results were in concordance with the number expected. They observed the short mutation length, and the long mutation length differed in the number of uninterrupted CGG repeats. The long uninterrupted CGG tract was correlated with the spread of the H3k9me3 repressive heterochromatin domain. They confirmed that local DNA methylation leads to a silent *FMR1* gene. They also discovered a large group of genes distal from *FMR1* that are repressed in FXS in concordance with H3K9me3 deposition. The distal genes have an important role in synaptic plasticity, testis development, and the reproductive system, which coincide with the phenotype of FXS [95].

Grosso et al. tried a different approach and combined short-read sequencing (Illumina) with long-read sequencing (Nanopore). They tried to combine the advantages of short-read and long-read sequencing. The approach is called indirect sequence capture. They started with a low quantity of DNA with high molecular weight, which is encapsulated in individual droplets and is used to amplify a sequence of 100–150 bp, which is near the region of interest. The molecules are recovered after encapsulation, and then long-read sequencing is performed to quantify the repeats, and short-read sequencing is used to search for intragenic variants in *FMR1*. Illumina generated 11,493,290 reads, with 462 coverage on *FMR1*, and Nanopore generated 170,532 reads with 357 coverage. Nanopore precisely quantified the number of repeats and interruptions and identified PM and FM. Then, Illumina sequencing was performed to search for point mutations or deletions, which had higher coverage than Nanopore [96].

In Table 1, we summarize the applications and the limitations of every technique previously described.

## 4. Conclusions

Fragile X syndrome is still a challenging diagnosis for current technologies. Most of the cases (98%) are the consequence of a high expansion number of CGG repeats, which leads to hypermethylation of the promoter and silencing of the *FMR1* gene.

For correct management of an FXS case, it is necessary to quantify the number of repeats, analyse the methylation status, identify the level of mosaicism, and detect the presence of AGG interruptions. For cases without large CGG repeats, the search for point mutations, deletions, or duplications is indicated.

The classic techniques can provide useful information about repeat numbers, methylation status, AGG interruptions, mosaicism, and PM status. Those techniques can also identify deletions, duplications, and point mutations, but the major drawback is the use of complementary techniques and the large amounts of DNA required. Optical genome mapping enables FXS diagnosis, but cannot precisely identify the number of repetitions, the methylation status, and interruptions.

Long-read sequencing presents the potential to fully replace classical techniques. The new technologies are capable of obtaining more information in a single assay. A complete characterisation of patients will pave the way for a better understanding and management of the disease and could lead to future therapies. Long-read sequencing is becoming more accessible, but future studies are needed for it to be optimized and used routinely.

## Figures and Tables

**Figure 1 ijms-24-09206-f001:**
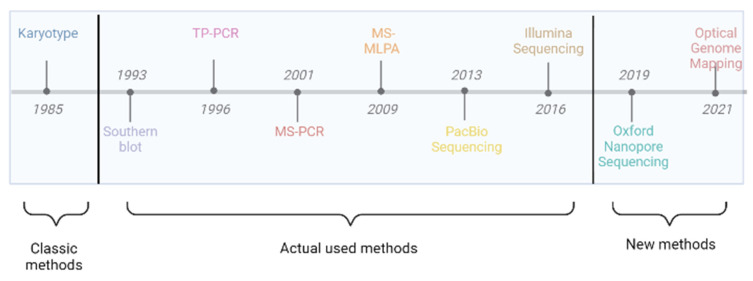
Techniques used for fragile X Syndrome diagnosis—Historical data: karyotype (fragile site in special conditions), Southern blot (identifies repeats), PCR (quantifies repeat number), MS-PCR and MS-MLPA (methylation status), long-read sequencing (combines all benefits), and optical genome mapping.

**Figure 2 ijms-24-09206-f002:**
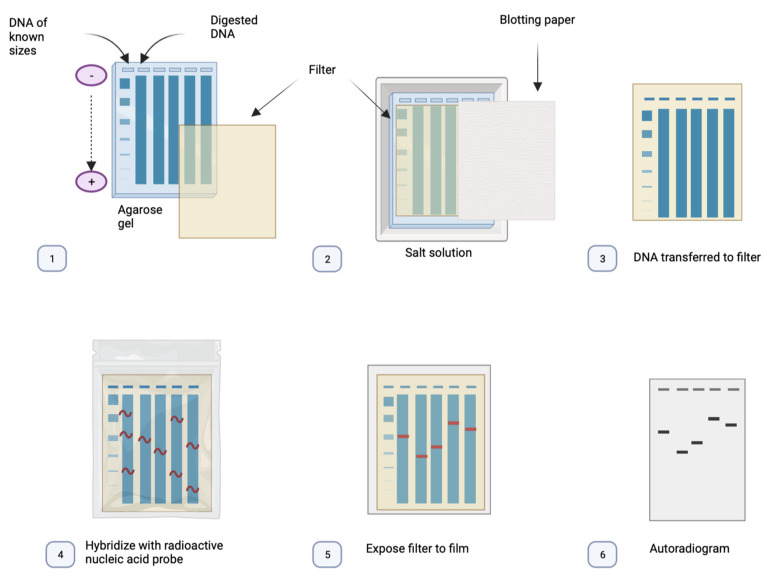
The base principles of Southern blot: (1) The DNA is fragmented using a digestion enzyme. (2) The DNA is denatured and separated by electrophoresis. (3) The DNA is transferred to a nitrocellulose filter (positively charged). (4) The DNA is marked using a labelled probe with the complementary sequence for the region of interest. (5) The filter is exposed to an X-ray film. (6) The DNA can be visualised as a pattern of bands on an autoradiogram.

**Figure 3 ijms-24-09206-f003:**
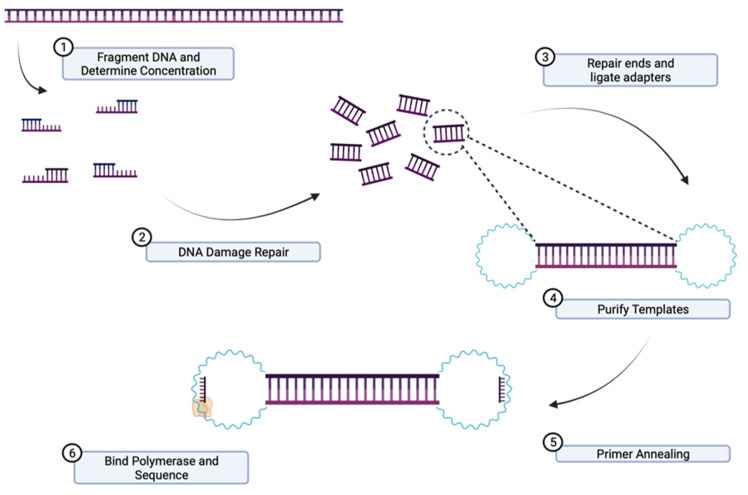
Principles of PacBio sequencing: (**1**) High-molecular-weight DNA (HMW DNA) is extracted, quantified, and fragmented. (**2**) DNA damage is repaired. (**3**) The ends of fragmented DNA are repaired, and hairpin adaptors are ligated on both ends of the double-stranded target DNA resulting in a circular molecule. (**4**) The template DNA is purified. (**5**) Primer annealing. (**6**) The template called SMRT-bell is loaded onto SMRT-cell and diffuses in a room called zero-mode waveguide where the adaptors bind to a polymerase immobilized at the bottom.

**Figure 4 ijms-24-09206-f004:**
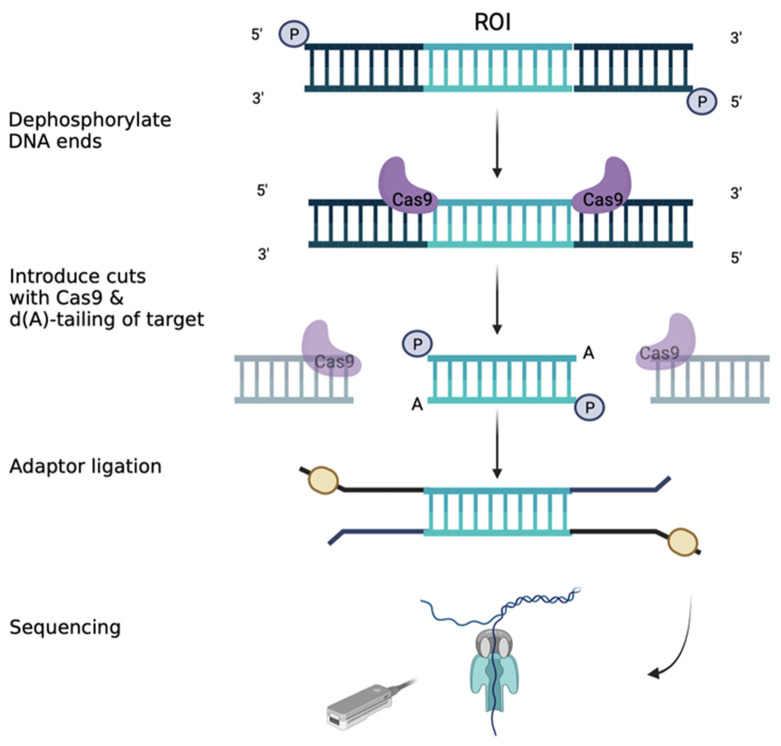
Library preparation using targeted Nanopore sequencing with Crispr-Cas9. The HMW DNA ends are dephosphorylated. Then the Cas9/guide RNA complexes introduce cuts around the region of interest (ROI). The RNA guides should target a nonrepetitive region and the specificity can be checked using diverse tools (e.g., Chop-Chop). The Nanopore adaptors are ligated around the ROI and the samples are loaded into the flow cell.

## Data Availability

Not applicable.

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
