# Peer review of "Narrative Review: Update on the Molecular Diagnosis of Fragile X Syndrome"

_ijms, 2023, doi:10.3390/ijms24119206_

Round 1
Reviewer 1 Report
Dear Author,
This manuscript is well written. The only flaw which I can point out in this paper is with very basic and well known information about the Fragile-X syndrome.
With no specific comments I would suggest for adding more literature with recent information/updates to get this manuscript published.
Dear Author,
The quality of English is good.
Author Response
Thank you for your valuable comments about our manuscript that helped us in the improvement of the paper.
Q1: This manuscript is well written. The only flaw which I can point out in this paper is with very basic and well known information about the Fragile-X syndrome. With no specific comments I would suggest for adding more literature with recent information/updates to get this manuscript published.
R1: The article was updated with more recent data, as was suggested. The main section where new literature has been added is the techniques section (Southern blot, Triplet Repeat Primed PCR, Digital PCR, Long Range Sequencing).

Reviewer 2 Report
In the manuscript entitled “Narrative review: update on molecular diagnostic of Fragile X Syndrome”, the authors review the molecular approaches in the identification of variations in the FMR1 (Fragile X Messenger Ribonucleoprotein) gene in patients with Fragile X syndrome (FXS). Comparing and contrasting the molecular techniques, they provide examples from the field and summarize different strategies by showing their weaknesses and strengths in different scenarios. Although the manuscript is informative with respect to the molecular genetics of fragile X syndrome, it does not present an original idea. The authors did not pay much attention to the writing, legends and referencing and the manuscript requires major revision.
1. Considering that this is a review paper, the references should be selected based on the original papers where discoveries were first made and main observations regarding fragile X syndrome was first published. However this is not the case. For example, introductory sentence of the manuscript defines the disorder and its genetic basis. However, the reference is a 2023 publication (Stone et al.) and not the original discovery paper that was published in the early 1990's and defined the field of diseases due to trinucleotide repeat expansion.
2. Other major reference problems do exist in the manuscript. Page 3, paragraph starting with “Fu et. al. were the first…”.
- The name given does not correspond to reference 13 in the reference list. Moreover, it should be Fu et al., not Fu et. al.
- Likewise, in the next paragraph, Yrigollen et al. corresponds to reference 23 in the reference list, but reference 23 was attributed to Nolin, Sarah L. et al. in the text. In the list, reference 15 corresponds to Nolin et al.
- This is a serious problem. Since inappropriate citing is an act of plagiarism, the authors must ensure the information in the manuscript is correctly attributed to their sources.
- Lastly, only the surname of the first author should be attributed in-text citations. For example, it is not Nolin, Sarah L. et al., but Nolin et al. This mistake was also made in the last paragraph of page 7. Should be Sitzmann et al., instead of Sitzmann, Adam F. et al.
3. Page 3, last paragraph. The claim that Southern blot can detect all FMR1 alleles is not really accurate. Southern blotting fails to capture variations caused by deletions or missense mutations. See the citation, which is also referenced in the text (Tassone F. (2015). Advanced technologies for the molecular diagnosis of fragile X syndrome. Expert review of molecular diagnostics, 15(11), 1465–1473.)
4. The legends for figures are far from being explanatory. They should be extended and properly labelled.
- Figure 1: The legend is too short to describe it. Abbreviations should be explained. If necessary, references could be given.
- Figure 2: Steps 1 and 2 lack captions. What are done in these steps? The legend should provide additional information than found on the illustration.
- Figure 3: The text in this legend is directly taken from the manuscript. It should provide some aspects that are not mentioned in the text.
- Figure 4: There is a label for it but there is no Figure 4 in the manuscript.
5. The conclusion part adequately summarizes the current technologies in the molecular diagnosis of FXS, but it lacks a paragraph that presents the concluding remarks on the trends, the capabilities of current technologies, and the future prospects in the diagnosis of FXS. They could also mention about technologies in development or the ones recently emerged in molecular diagnostics and discuss their potential use in the identification of FXS.
6. Advantages and disadvantages of each diagnostic technique are implicated in the text but not in the table. This type of comparison table should be made. Current version of manuscript is lacking from this precious information.
Minor comments:
- Page 3, paragraph 5 starting with “FMRP is an RNA binding protein…”.
- Line 5. Not ribosomes translation, but ribosomal translation.
- Line 6. Not glutamaergic, but glutamatergic.
- Page 4, section 3.1. Karyotype.
- It would be better to call karyotyping or karyotype analysis. Karyotype term is mostly used to refer to the depiction of chromosomes, rather than a name for a technique. Plus, there is no need for capitalizing it as in the first line.
- Line 4. It should be “difficulty in identifying”, but not “difficulty to identify”.
- In formal conversations, contractions like “can’t, doesn’t, didn’t” should be avoided. They have been used many times throughout the paper.
- Page 6, paragraph 3, line 3. Not “proceeds two PCR reactions” but “proceeds with two PCR reactions”.
- Page 6 & 7, section 3.7. Optical Genome Mapping. The commercial name for the system is not Saphyre but Saphyr.
- Page 9, paragraph 4. The year after citation should be taken into parentheses. Therefore, it is not Payne et al. 2021 but Payne et al. (2021).
- Page 10, paragraph 1, line 8. The first letter of protein names should be capitalized. Therefore, it is not cas9, but Cas9.
- Page 10-11. Fitting Table 1 in a single page would improve readability.
- Throughout the text, double spaces are used after full stops at multiple occurrences.
- In abstract, the first sentence should include abbreviation of Fragile X syndrome which is FXS.
- Table 1 is likely to be summary of the article. Therefore, its design and feature columns should be improved. In terms of design, for example, “Interrup-tion” “Diag-nosis” can be written as single word without “-“ usage.
- Page 3, “…where four guanines are linked by Hoogsteen hydrogen” needs additional information about Hoogsteen hydrogen by adding “which is …."
In general, the quality of English is acceptable. But moderate revision is advised. Please see the comments section.
Author Response
Thank you for your valuable comments about our manuscript that helped us in the improvement of the paper.
Q1: 1. Considering that this is a review paper, the references should be selected based on the original papers where discoveries were first made and main observations regarding fragile X syndrome was first published. However, this is not the case. For example, introductory sentence of the manuscript defines the disorder and its genetic basis. However, the reference is a 2023 publication (Stone et al.) and not the original discovery paper that was published in the early 1990's and defined the field of diseases due to trinucleotide repeat expansion.
R1: The manuscript was updated by adding references that are related to the original works.
Q2: Other major reference problems do exist in the manuscript. Page 3, paragraph starting with “Fu et. al. were the first…”.
- The name given does not correspond to reference 13 in the reference list. Moreover, it should be Fu et al., not Fu et. al.
- Likewise, in the next paragraph, Yrigollen et al. corresponds to reference 23 in the reference list, but reference 23 was attributed to Nolin, Sarah L. et al. in the text. In the list, reference 15 corresponds to Nolin et al.
- This is a serious problem. Since inappropriate citing is an act of plagiarism, the authors must ensure the information in the manuscript is correctly attributed to their sources.
- Lastly, only the surname of the first author should be attributed in-text citations. For example, it is not Nolin, Sarah L. et al., but Nolin et al. This mistake was also made in the last paragraph of page 7. Should be Sitzmann et al., instead of Sitzmann, Adam F. et al.
R2: The erroneous citations into the manuscript were corrected.
Q3: Page 3, last paragraph. The claim that Southern blot can detect all FMR1 alleles is not really accurate. Southern blotting fails to capture variations caused by deletions or missense mutations. See the citation, which is also referenced in the text (Tassone F. (2015). Advanced technologies for the molecular diagnosis of fragile X syndrome. Expert review of molecular diagnostics, 15(11), 1465–1473.)
R3: The text was updated in order to make it more clearer and information was added to stress the importance and pitfalls of the Southern Blot technique in FXS diagnostic.
R4: The legends for figures are far from being explanatory. They should be extended and properly labelled.
- Figure 1: The legend is too short to describe it. Abbreviations should be explained. If necessary, references could be given.
- Figure 2: Steps 1 and 2 lack captions. What are done in these steps? The legend should provide additional information than found on the illustration.
- Figure 3: The text in this legend is directly taken from the manuscript. It should provide some aspects that are not mentioned in the text.
- Figure 4: There is a label for it but there is no Figure 4 in the manuscript.
Q4: All figures’ captions were rewritten. Figure 4 was reintroduced in the main text.
Q5: The conclusion part adequately summarizes the current technologies in the molecular diagnosis of FXS, but it lacks a paragraph that presents the concluding remarks on the trends, the capabilities of current technologies, and the future prospects in the diagnosis of FXS. They could also mention about technologies in development or the ones recently emerged in molecular diagnostics and discuss their potential use in the identification of FXS.
R5: A conclusion about the future use and utility of different techniques was added.
Q6: Advantages and disadvantages of each diagnostic technique are implicated in the text but not in the table. This type of comparison table should be made. Current version of manuscript is lacking from this precious information.
R6: The table was supplemented with two more columns in which are presented the advantages and disadvantages of the main diagnostic methods.
Q7: Minor comments:
- Page 3, paragraph 5 starting with “FMRP is an RNA binding protein…”.
- Line 5. Not ribosomes translation, but ribosomal translation.
- Line 6. Not glutamaergic, but glutamatergic.
- Page 4, section 3.1. Karyotype.
- It would be better to call karyotyping or karyotype analysis. Karyotype term is mostly used to refer to the depiction of chromosomes, rather than a name for a technique. Plus, there is no need for capitalizing it as in the first line.
- Line 4. It should be “difficulty in identifying”, but not “difficulty to identify”.
- In formal conversations, contractions like “can’t, doesn’t, didn’t” should be avoided. They have been used many times throughout the paper.
- Page 6, paragraph 3, line 3. Not “proceeds two PCR reactions” but “proceeds with two PCR reactions”.
- Page 6 & 7, section 3.7. Optical Genome Mapping. The commercial name for the system is not Saphyre but Saphyr.
- Page 9, paragraph 4. The year after citation should be taken into parentheses. Therefore, it is not Payne et al. 2021 but Payne et al. (2021).
- Page 10, paragraph 1, line 8. The first letter of protein names should be capitalized. Therefore, it is not cas9, but Cas9.
- Page 10-11. Fitting Table 1 in a single page would improve readability.
- Throughout the text, double spaces are used after full stops at multiple occurrences.
- In abstract, the first sentence should include abbreviation of Fragile X syndrome which is FXS.
- Table 1 is likely to be summary of the article. Therefore, its design and feature columns should be improved. In terms of design, for example, “Interrup-tion” “Diag-nosis” can be written as single word without “-“ usage.
- Page 3, “…where four guanines are linked by Hoogsteen hydrogen” needs additional information about Hoogsteen hydrogen by adding “which is …."
R7: All the suggested modifications were included in this revised version of the manuscript.

Round 2
Reviewer 1 Report
Dear Author/s,
The manuscript is having the potential to get published.
English is of good quality.